

# Functional group based marine ecosystem assessment for the Bay of Biscay via elasticity analysis

Geoffrey R. Hosack[1] and Verena M. Trenkel[2]

[1] Commonwealth Scientific and Industrial Research Organisation, Data61, Hobart, Tasmania, Australia
[2] Ifremer, Nantes, France

## ABSTRACT

The transitory and long-term elasticities of the Bay of Biscay ecosystem to density-independent and density-dependent influences were estimated within a state space model that accounted for both process and observation uncertainties. A functional group based model for the Bay of Biscay fish ecosystem was fit to time series obtained from scientific survey and commercial catch and effort data. The observation model parameters correspond to the unknown catchabilities and observation error variances that vary across the commercial fisheries and fishery-independent scientific surveys. The process model used a Gompertz form of density dependence, which is commonly used for the analysis of multivariate ecological time series, with unknown time-varying fishing mortalities. Elasticity analysis showed that the process model parameters are directly interpretable in terms of one-year look-ahead prediction elasticities, which measure the proportional response of a functional group in the next year given a proportional change to a variable or parameter in the current year. The density dependent parameters were also shown to define the elasticities of the long term means or quantiles of the functional groups to changes in fishing pressure. Evidence for the importance of indirect effects, mediated by density dependence, in determining the ecosystem response of the Bay of Biscay to changes in fishing pressure is presented. The state space model performed favourably in an assessment of model adequacy that compared observations of catch per unit effort against cross-validation predictive densities blocked by year.

## INTRODUCTION

Functional groups provide a structured and quantifiable set of indicators that enable monitoring trends of ecosystem state and predictions of ecosystem response to anthropogenic and environmental pressures (*Hayes et al., 2015*). Functional groups are a relevant level of biological organisation to investigate community stability (*Bell, Fogarty & Collie, 2014*) and restructuring (*Frank et al., 2011*), interactions with human activities (*Griffith & Fulton, 2014*) including cumulative impacts of fishing pressure (*Kaplan, Gray & Levin, 2013*), evaluation of protected area efficacy (*Strain et*

Corresponding author
Geoffrey R. Hosack,
geoff.hosack@csiro.au

*al., 2019*) and mitigation for climate change induced impacts (*Cinner et al., 2009*). The prediction and trend estimation of indicators within complex ecosystems requires the development of modelling techniques that coherently summarise empirical data, expert knowledge and uncertainty (*Luo et al., 2011*). Understanding is further facilitated if the estimated model parameters correspond to interpretable summaries of ecosystem dynamics.

Various models have been used for determining functional group ecosystem structure, the strength of their ecological interactions, and the capacity for indirect effects under fisheries and environmental pressures (*Bell, Fogarty & Collie, 2014*; *Gaichas et al., 2017*). There are two general strategies for functional group modelling. The first strategy consists of modelling population dynamics of individual species, and then grouping species into functional groups for analysis. The second strategy models the dynamics of functional groups directly (e.g., *Plagányi et al., 2014*). An example of the first strategy is *Gaichas et al. (2017)*, who used a length-based ten species model, proposed by *Hall et al. (2006)* and further developed by *Rochet et al. (2011)*, to investigate functional group level quota setting and trade-offs between fishing gears. As an example of the second strategy, *Lucey et al. (2012)* estimated maximum sustainable yield by fitting surplus production models to functional groups in a dozen ecosystems. As another example of the second strategy, a functional group level production model that allowed for interactions among groups was fitted by *Bell, Fogarty & Collie (2014)* to survey time series from 19 exploited marine ecosystems. Including inter-functional group dependence elucidated the degree of intra- and inter-group density dependence and hence the potential for indirect effects.

In this study, we chose the second strategy to study the functional group dynamics in the Bay of Biscay marine ecosystem and evaluate the evidence for inter-functional group density dependence in recent decades. For further exploring the capacity for long-term cumulative impacts of fishing pressure, we carry out scenario projections of extreme fisheries management changes, including no fishing on small pelagics or no bottom trawling. The functional groups considered are the same as in *Bell, Fogarty & Collie (2014)*. The model is a multi-group Gompertz model similar to *Bell, Fogarty & Collie (2014)*, but considers fishing mortality rates as unknown (c.f. *Bell, Fogarty & Collie, 2014* who include annual functional group fishing pressure as a known quantity). The Gompertz model examined in this study allocates equal prior probability to bottom-up trophic effects and top-down control by predation. This is accomplished by setting identical prior distributions on the magnitudes of direct top-down predator effects on prey and direct bottom-up prey effects on predators.

Two broad approaches are generally used for parameterising marine food web models. In the first approach parameterisation is carried out using a combination of values derived from data, the literature or ecological theory (*Lindstrøm, Planque & Subbey, 2017*). In addition, a small number of model parameters are sometimes tuned jointly, for example to achieve biomass balance in an Ecopath model (e.g., EwE model for Bay of Biscay, *Lassalle et al., 2011*). In this approach, it is difficult to quantify the uncertainty associated with the final reported values of the model parameters. In the second approach, all model parameters as well as all state variables such as biomass for each functional group are estimated jointly and thus coherently. For complex simulation models, various statistical methods based on

likelihood approximations and summary statistics have been proposed for this task (see review in *Hartig et al., 2011*).

For models with a tractable likelihood function, such as relatively simple food web or multi-species models, joint estimation of unknown process and observational quantities can be carried out in a state space framework, using maximum likelihood (ML; e.g., *Spencer & Ianelli, 2005*) or a Bayesian approach (e.g., *Hosack, Trenkel & Dambacher, 2013*; *Hosack, Peters & Ludsin, 2014*). In contrast, stock assessments often model species and population dynamics with greater detail, and in these analyses the parameters are typically not jointly estimated in a Bayesian framework (*Aeberhard, Flemming & Nielsen, 2018*). Whether a ML or a Bayesian approach is chosen, the important point is being able to coherently consider all sources of uncertainty while accurately and clearly documenting model assumptions. Indeed, when using models to draw conclusions about the state of ecosystems, it is important to acknowledge that no ecological model perfectly represents the real world, which makes the explicit inclusion of sources of uncertainty essential (*Ruiz & Kuikka, 2012*). The state space framework has the advantage of explicitly modelling both biological process uncertainty and observation uncertainties. We therefore formulated the food web model in state space form and fitted it to time series data using a Bayesian approach.

The Bayesian state space model jointly estimates the basic parameters of the multivariable fisheries model. In this paper, a stochastic discrete-time multivariate model is used. The process model approximates a multivariate generalisation of the Fox surplus production model, which uses the Gompertz form of density dependence (*Fox, 1970*). This multivariate process model has been advocated as useful for monitoring trends and responses of indicators to anthropogenic and environmental pressures in marine ecosystems (*Torres et al., 2017*). Including observation error results in a linear state space model for the unobserved latent biomass or abundance after logarithmic transformation (*Thompson, 1996*). The extension of this state space model to multiple latent trajectories that represent the unobserved true abundances of populations, species or functional groups is a popular choice for the analysis of multivariate time series in aquatic ecology (*Ives et al., 2003*; *Spencer & Ianelli, 2005*; *Lindegren et al., 2009*; *Hosack, Trenkel & Dambacher, 2013*; *Bell, Fogarty & Collie, 2014*) and elsewhere in the ecological and environmental domains (*Luo et al., 2011*). For a single population, it has been shown that incorporation of multiple observational time series without aggregation or pooling results in better estimates of the Gompertz model parameters (*Dennis, Ponciano & Taper, 2010*). Therefore this study additionally incorporates the multiple observational times series that are available for each functional group in the Bay of Biscay ecosystem.

The Bay of Biscay ecosystem is subject to multiple pressures (*Lorance et al., 2009*) and home to a diversity of fishing fleets (*Daurès et al., 2009*). The structure of the food web has changed in recent decades, driven by environmental conditions and a reduction in overall fishing pressure (*Rochet, Daurès & Trenkel, 2012*; *Rochet, Collie & Trenkel, 2013*). The number of fishing vessels has continuously decreased since the 1950s after repeated vessel decommissioning and fleet capacity reduction programs, and fishing power has decreased from the 1990s (*Mesnil, 2008*). A massive vessel payback program in the early 2000s (*EU, 2006*) accelerated the decrease in French fishing power in the Bay of

Biscay (*Rochet, Daurès & Trenkel, 2012*). In reaction to poor recruitment, anchovy fishing was banned between 2005 and 2009, forcing dependent fisheries to look for alternative fishing opportunities (*Andres & Prellezo, 2012*). For the Bay of Biscay ecosystem, both quantitative (Ecopath) and qualitative modelling results identified benthivores and planktivores (small pelagics) as the groups most sensitive to pressure changes (*Lassalle et al., 2014*). Using an Ecopath with Ecosim model, the importance of small pelagics for food web structure was confirmed by simulations with reduced fishing levels for these species (*Moullec et al., 2017*). *Rochet, Collie & Trenkel (2013)* found empirical evidence for compensation within the planktivore functional group, but synchrony for piscivores. It was suggested that this result could be explained by fisheries targeting a broader range of demersal species. In contrast, planktivore dynamics are generally more driven by environmental variations (reviewed by *Trenkel et al., 2014*).

In this study, a functional group based Gompertz model was fit to time series obtained from scientific survey and commercial catch and effort data from the Bay of Biscay ecosystem. Previous multivariate Gompertz models for marine ecosystems with explicit incorporation of fishing pressure have assumed known fishing mortalities (*Lindegren et al., 2009*; *Bell, Fogarty & Collie, 2014*; *Torres et al., 2017*) or approximated annual harvested fractions with an instantaneous rate (e.g., *Spencer & Ianelli, 2005*), which results in a linear model amenable to closed form estimation of the latent states and likelihood (i.e., the Kalman filter, *Harvey, 1989*). For the Bay of Biscay ecosystem model, fishing mortality is time-varying and considered as unknown and instantaneous in the process model. In the observation model, the instantaneous fishing rates are related to the observed annual catch and harvested fraction, which results in a non-linear state estimation problem for the unknown fishing rates (or fractions). Estimation instead proceeds by Monte Carlo methods applied within a Bayesian perspective. The observations include both fishery independent survey data and commercial catch and effort data. The estimated unknown quantities include the magnitude of observation and process error, the fishing mortality rates, and both density independent and density dependent growth parameters. Given the model assumptions documented by the prior distribution and the observational data, the posterior estimates of the unknown process parameters, unobserved latent biomasses of the functional groups and functions of these quantities become available.

The parameters of the Gompertz model are shown to be directly interpretable as elasticities that evaluate ecosystem response to perturbations. Elasticities originated in economics (*Hicks, 1946*) and are important in population biology (*Caswell, 2001*). The parameters of the discrete-time multivariate Gompertz model are shown to be directly interpretable as the one-year look-ahead prediction elasticities. These elasticities are defined as the proportional change in the biomass one year in the future given a proportional change to a variable or parameter in the current year. At time scales comparable to the duration of the observation period, the predictive capacity of the transitory elasticities that correspond to one-year ahead predictions may be assessed. Model adequacy is therefore assessed using cross-validation predictive densities (*Gelfand, 1996*) that successively leave out all the catch per unit effort data in each year of the observation period. In the long-term, the density dependent parameters are also shown to define the long-term elasticities of functional

group means or quantiles to a sustained change in fishing pressure. The implications of cumulative impacts of fishing pressure (e.g., *Kaplan, Gray & Levin, 2013*; *Link & Marshak, 2019*; *Zhou et al., in press*) and long-term management decisions to sustained reduction of fishing effort on certain gear and functional group combinations are investigated using the estimated long-term elasticities.

## DATA

Fishing fleets were grouped into broad gear categories similar to *Trenkel et al. (2013)*, as were the functional groups (Table S1). Empirical analyses have shown that each fleet only impacts and depends economically on a small number of species (*Daurès et al., 2009*) and functional groups (*Trenkel et al., 2013*). A French fishery (gear type) was deemed to target a functional group if the fisheries contribution to the total landings of a given functional group was more than 10% and the functional group contributed more than 10% to the landings of that fishery for the years 2000–2015. The analysis was further restricted to fishing gear operated from vessels. Figure 1 shows the non-negligible relationships between targeted fisheries and the functional groups.

The following data sets were used in this study. (1) Survey biomass density (kg per km$^2$) averaged across hauls for the period 2000–2015 for pelagic planktivores ($K$), benthivores ($B$) and demersal piscivores ($D$). Survey data are for all observed species, independent of whether they are landed by commercial fisheries or not (some species are entirely discarded). The survey used a Grande Ouverture Verticale (GOV) bottom trawl and a stratified random sampling design (*Poulard & Trenkel, 2007*); for this study the stratification was not considered. The bottom trawl survey was assumed to target all functional groups except pelagic piscivores. (2) Annual international landings (metric tonnes) in the Bay of Biscay (ICES divisions 8a and 8b) for the period 1950–2015 (*ICES, 2011*; *ICES, 2017*) for all functional groups: $B$, $D$, $K$, pelagic piscivores ($G$). Diadromous fish species were removed as they spend only part of their time on the continental shelf. The data contained all landed species, including those not caught by the bottom trawl survey. (3) Annual French fishing landings (kg) and effort data for the Bay of Biscay provided by the French administration (Direction des pêches maritimes et de l'aquaculture) for the period 2000–2015; effort data were days-fishing by gear (dredgers, hooks, mixed trawlers, netters, pelagic trawlers, potters, and purse seiners). Mixed trawlers included different types of bottom trawls.

For the analysis, international landings were converted to kilotonnes (1,000 metric tonnes), survey catch per unit effort (CPUE) was converted to metric tonnes per km$^2$ and French commercial fishery CPUE was converted to metric tonnes divided by days fishing. The international landings data for ICES division 8a and 8b are given in Fig. S1 and the CPUE data used in the analyses are plotted in Fig. S2.

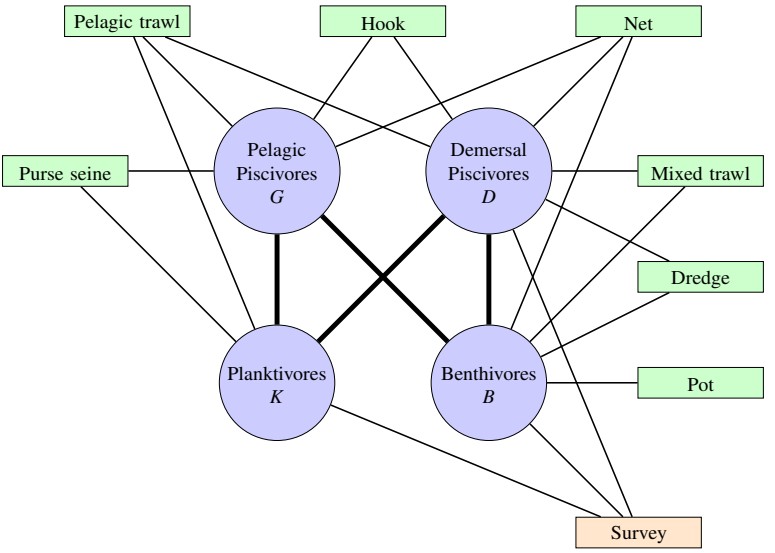

**Figure 1 Functional groups (blue circles), fisheries (green blocks) and fishery independent scientific surveys (orange block) of the Bay of Biscay ecosystem.** Pathways show trophic connections and targeted combinations of gear type and functional groups.

## MODEL

### Observation model

Conditional on the biomass $x_i(t)$ of the $i$th functional group in year $t$, the observation model for the French commercial fishery and fishery independent survey CPUE data is given by,

$$p(C_{ik}(t)|x_i(t), E_k(t), q_{ik}, \sigma_k) = LN\left(C_{ik}(t)|\log(q_{ik}E_k(t)x_i(t)), \sigma_k^2\right), \qquad (1)$$

where $LN(\cdot|m, s^2)$ is a log-normal distribution with mean $m$ and variance $s^2$ on the log-scale. $C_{ik}(t)$ is the catch in metric tonnes for functional group $i \in \{K, B, D, G\}$ in year $t$ by gear type $k \in \{Dredge, Hook, Mixed\ trawl, Net, Pelagic\ trawl, Pot, Purse\ seine, Survey\}$ with catchability $q_{ik}$ and effort $E_k(t)$. The observation error is given by $\omega_{ik}(t) \overset{i.i.d.}{\sim} N(0, \sigma_k^2)$ for $i \in \{K, B, D, G\}$ and the $k$th gear type. Equation (1) applies to the targeted combinations of gear type described above and as shown in Fig. 1. The catches from incidental combinations of gear type and functional group are assumed ignorable and not included in the likelihood.

The impact of catch from French landings on the unobserved latent state of biomass for the $i$th functional group in year $t$ is captured by the international landings $h_i(t)$. The catch component $C_{ik}(t)$ of the observed CPUE from French fisheries thus explicitly enters only the observation model of Eq. (1). With no error in international reported landings, the fished fraction for the $i$th functional group in year $t$ is given by,

$$F_i(t) = \frac{h_i(t)}{x_i(t)} = 1 - e^{-f_i(t)}, \qquad (2)$$

where $f_i(t)$ is the instantaneous rate of fishing mortality on the $i$th functional group in year $t$. However, it is possible for errors or misreporting of landings to occur. Uncertainty was

allowed for by specifying a lognormal distribution for annual international landings such that $h_i(t) \sim LN(\log F_i(t) + \log x_i(t), 0.001)$, which approximately provides probability 0.9 of less than 5% reporting error.

The scientific survey data for the relevant functional groups (i.e., excluding pelagic piscivores) is similarly addressed as with the commercial CPUE data, but with different units: average metric tonnes of biomass per square kilometre for each functional group by survey year. The observation error has independent unknown variance, which reflects the fact that the observations from the scientific surveys align to an unknown degree with the fishable biomass of a given functional group.

## Process model

*Fox (1970)* introduced the continuous-time Gompertz-Fox model for a fish population $x(t)$ given constant effort $E$ and catchability $q$ with the ordinary differential equation

$$\frac{dx}{dt} = bx \log \frac{\beta}{x} - fx, \tag{3}$$

where $b$ is related to the density independent growth rate at a particular population level (see *Thompson, 1996*), $\beta$ is the carrying capacity of the environment and $f = qE$ is the instantaneous fishing mortality rate. The first term of Eq. (3) is the Gompertz form of density dependence (*Thieme, 2003*). Extension of Eq. (3) to multiple variables changes the interpretation of the Gompertz-Fox model parameters because the carrying capacity for the $i$th variable, which is simply $\beta$ in Eq. (3), may now depend on other variables. Consider the multivariate extension of the Gompertz-Fox model to the $n$-dimensional system

$$\frac{dx}{dt} = \text{diag}[x] (r - f + A \log x), \tag{4}$$

where $f$ is the $n$-dimensional vector of instantaneous fishing mortality rates and $A$ is the $n \times n$ matrix of density-dependent parameters. The $n$-dimensional vector of density independent growth rates $r$ gives the per capita growth rates in the absence of fishing or density dependence. If the variables of the above equation are independent such that $A$ is a diagonal matrix, then the substitutions $r_i = b_i \log \beta_i$ and $a_{ii} = -b_i$ recover the parameterisation of Eq. (3) for variable $x_i$.

Different choices of discretisation methods applied to Eq. (3) will lead to different discrete-time models (e.g., *Turchin, 2003*), and so also for multivariate models. A deterministic multivariate discrete-time version of Eq. (4) is given by

$$x(t+1) = \text{diag}[x(t)] \exp [r - f + A \log x(t)], \tag{5}$$

where, as for Eq. (4), $r$ is the vector of density independent growth parameters, $f$ is the vector of instantaneous fishing mortalities and the matrix $A$ determines the strength of density-dependence within and among variables.

The discrete-time model of Eq. (5) shares a key property associated with the long term behaviour of the continuous-time model given by Eq. (4). For small time steps in Eq. (5), the biomass of the $i$th variable at time $t + \tau$ is

$$x_i(t+\tau) = x_i(t) \exp \left[ \tau \left( r_i - f_i + \sum_j A_{ij} \log x_j(t) \right) \right].$$

As the time step $\tau$ approaches zero,

$$\frac{dx_i}{dt} = \lim_{\tau \to 0} \frac{x_i(t+\tau) - x_i(t)}{\tau} = x_i(t) \lim_{\tau \to 0} \frac{\exp\left(r_i - f_i + \sum_j A_{ij} \log x_j(t)\right)^{\tau} - 1}{\tau}$$

$$= x_i(t) \left[ r_i - f_i + \sum_j A_{ij} \log x_j(t) \right]. \tag{6}$$

Equation (6) is equivalent to the continuous-time model given by Eq. (4). Note that if $r_i - f_i + \sum_j A_{ij} \log x_j(t) = 0$ then both $dx_i/dt = 0$ in Eq. (4) and $x_i(t+\tau)/x_i(t) = 1$ in Eq. (5). The continuous-time model of Eq. (4) and the discrete-time model of Eq. (5) therefore share the same interior equilibrium given by $x^* = \exp\left[-A^{-1}(r - f)\right]$. In this sense, Eq. (5) is an interpretable discretisation of the Fox model extended to a multidimensional system because its parameters have the same functions as those of Eq. (4).

### Bay of Biscay

A stochastic dynamic ecosystem model based on Eq. (5) is developed for the Bay of Biscay food web (Fig. 1). Let the functional groups be identified by $K = Planktivore$, $B = Benthivore$, $D = Demersal\ piscivore$, and $G = Pelagic\ piscivore$ and the functional group biomasses at time $t$ by $x(t) = [x_K(t), x_B(t), x_D(t), x_G(t)]^\top$. Here, the top carnivores are pelagic and demersal piscivores. Planktivores and benthivores are the basal resources. An entry $A_{ij}$ of the matrix

$$A = \begin{array}{c} \\ K \\ B \\ D \\ G \end{array}\begin{array}{c} \begin{array}{cccc} K & B & D & G \end{array} \\ \left(\begin{array}{cccc} -a_{KK} & 0 & -a_{KD} & -a_{KG} \\ 0 & -a_{BB} & -a_{BD} & -a_{BG} \\ a_{DK} & a_{DB} & -a_{DD} & 0 \\ a_{GK} & a_{GB} & 0 & -a_{GG} \end{array}\right) \end{array},$$

determines the sign and magnitude $a_{ij}$ of the rate of biomass transfer from functional group $i$ to functional group $j$ in the Bay of Biscay. A negative (positive) sign indicates a negative (positive) density dependent relationship, where the per capita growth rate $g_i(x(t)) = x_i(t+1)/x_i(t)$ of the $i$th functional group decreases (increases) as the biomass of the $j$th functional group increases. The qualitative sign structure of the matrix $A$ specifies that the consumer functional groups $D$ and $G$ exert negative density dependence (top-down control) by predation pressure on the basal resources $K$ and $B$. The basal resources exert positive density dependence (bottom-up effects) by providing energy for the consumer functional groups. Intra-functional group negative density dependence is induced by interactions among and within species in each of the four compartments.

As above, let the annual harvest of functional group $i$ as reported from recorded international landings for ICES divisions 8a and 8b (Bay of Biscay) in year $t$ be denoted by $h_i(t)$, $i \in \{K, B, D, G\}$. The recorded annual harvest $h_i(t)$ for functional group $i$ in year $t$ is considered in units of one thousand metric tonnes or kilotonnes (kt). The biomass variables $x(t)$ inherit these units and are thus defined in terms of fishable (not total) biomass. The instantaneous fishing mortality is allowed to vary piecewise constant by year, and from Eq.

(2) is given by

$$f_i(t) = -\log(1 - F_i(t)).\tag{7}$$

Process uncertainty or noise is modelled by a multiplicative process, $\epsilon(t)$, distributed multivariate normal on the log scale with mean zero and diagonal covariance matrix, $S = \text{diag}\left[\eta_K^2, \eta_B^2, \eta_D^2, \eta_G^2\right]$. From Eqs. (5) and (7), the ratio of biomass in year $t+1$ to the biomass in year $t$, which is equivalent to the per capita growth rate, is with process uncertainty given by,

$$g(x(t)) = \text{diag}[x(t)]^{-1}x(t+1) = \exp\left[r - f(t) + A\log x(t) + \epsilon(t)\right],\tag{8}$$

where $\epsilon(t) \overset{i.i.d.}{\sim} N(0, S)$. The (logarithmic) process error variance $\eta_i^2$ describes the magnitude of the uncertainty of the process dynamics for the functional groups. The transition density for biomass from year $t$ to year $t+1$ is a multivariate lognormal density,

$$p(x(t+1)|x(t), h(t), r, A, S) = LN(x(t+1)|\delta(t), S),\tag{9}$$

with location vector $\delta(t) = \log x(t) + r - f(t) + A\log x(t)$ and scale matrix $S$.

## ANALYSIS

### Elasticities for one year look-ahead predictions

In economics, elasticity is defined as the proportional response of a variable of interest to a proportional increase of another variable (*Hicks, 1946*, Ch. 16). Elasticities are also important in population biology (*Caswell, 2001*). Elasticities are useful because the focus on proportional change results in a non-dimensional analysis, which provides a common scale to support comparison of how the variable of interest responds to changes in parameters having different units. A fundamental quantity of interest is how the biomass of the $i$th functional group $x_i(t+1)$ proportionally responds to a proportional change in a variable $v(t)$ during the preceding year $t$. This quantity is the elasticity for a one-year look-ahead prediction given by

$$\frac{\partial x_i(t+1)/x_i(t+1)}{\partial v(t)/v(t)} = \frac{\partial \log x_i(t+1)}{\partial \log v(t)},$$

where $x_i(t+1)$ is the growth equation defined by Eq. (9). The above equation is an example of a transient elasticity (*Caswell, 2007*), here applied to a stochastic dynamic model. For a small perturbation of $v(t)$, an elasticity of $+1$ predicts that the next year's biomass will increase by the same percentage as the percentage increase in $v(t)$. If the elasticity is greater than $+1$ then the next year's biomass will increase by a greater percentage, and if the elasticity is less than $+1$ (but greater than zero) then the next year's biomass will increase by a lesser percentage. Negative elasticities are similarly interpreted but instead predict decreases in the next year's biomass given a small increase to $v(t)$.

From Eqs. (8) and (9), the logarithm of the growth equation for functional group $i$ from year $t$ to $t+1$ is given by

$$\log x_i(t+1) = (1 + A_{ii})\log x_i(t) + \sum_{j \neq i} A_{ij}\log x_j(t) + r_i - f_i(t) + \epsilon_i(t),$$

where $A_{ij}$ accounts for both the sign and the magnitude $a_{ij}$ of the density dependent effect of functional group $j$ on functional group $i$. The elasticity of the one-year look-ahead prediction for the biomass of the $i$th functional group in year $t+1$ given a change to the biomass of the $j$th functional group in year $t$ is then

$$\frac{\partial \log x_i(t+1)}{\partial \log x_j(t)} = \begin{cases} 1+A_{ii}=1-a_{ii} & \text{if } i=j \\ A_{ij}=\text{sgn}(A_{ij})a_{ij} & \text{if } i \neq j, \end{cases} \tag{10}$$

where $\text{sgn}(z)$ is $+1$, $0$ or $-1$ depending on whether the argument $z$ is positive, zero or negative. Note that the density dependent elasticities for the one-year look-ahead predictions are invariant over time.

Elasticities are defined for positive quantities. The density independent growth rate $r_i$ may however be negative, in which case it instead describes the rate of decay. For example, the growth of a consumer functional group may solely depend on the biomass of a producer functional group. Without its prey or any alternative sources of growth or mortality aside from density independence, then only a proportion $\exp[r_i] < 1$ of the consumer biomass will survive into the next year. To allow for negative $r_i$, the elasticity of biomass to a change in the magnitude $|r_i|$ is instead evaluated by substituting $\text{sgn}(r_i)|r_i|$ for $r_i$ in the logarithm of the growth equation $\log x_i(t+1)$. The elasticity of the one-year look-ahead prediction for a change in the magnitude of a non-zero density independent growth (decay) rate $r_i$ is then given by

$$\frac{\partial \log x_i(t+1)}{\partial \log |r_i|} = \frac{\partial x_i(t+1)}{\partial |r_i|} \frac{|r_i|}{x_i(t+1)} = x_i(t+1)\text{sgn}(r_i)\frac{|r_i|}{x_i(t+1)} = \text{sgn}(r_i)|r_i| = r_i,$$

which is invariant over time. In contrast, the elasticities of the one-year look-head predictions for the instantaneous fishing mortality rates vary by year,

$$\frac{\partial \log x_i(t+1)}{\partial \log f_i(t)} = -f_i(t),$$

where $f_i(t) > 0$.

## Long term dynamics

Consider the long-term implications of a management policy that holds the instantaneous fishing mortality constant for each functional group such that $f(t)=f$ for all $t$. This policy is equivalent to specifying a constant annual fished fraction of biomass, see Eq. (2). The logarithmically transformed dynamic system of Eq. (9) with time-invariant fishing mortalities is given by,

$$\log x(t+1)=(I+A)\log x(t)+r-f+\epsilon(t). \tag{11}$$

A dynamic system is strictly stationary if realisations of the process for a set of times $\tau = \{t_1, t_2, \ldots, t_T\}$ have the same joint probability distribution as realisations at times $\tau + s$ for constant $s$. The dynamic system of Eq. (11) is strictly stationary if the spectral radius of the matrix $I+A$ is less than one. Given stationarity, the joint distribution for $\log x(t)$ is multivariate normal with location vector $m = -A^{-1}(r-f)$ and covariance matrix $P$ that solves $P=(I+A)P(I+A)^\top+S$ (*Harvey, 1989*, Ch. 3). The distribution for $x(t)$ is

therefore multivariate lognormal, $x(t) \sim LN(m, P)$, such that $\hat{x} = \exp[m]$ is the vector of long-term median biomasses of the functional groups and $\bar{x} = \exp\{m + \mathrm{diag}[P]/2\}$ gives the long-term average biomasses.

## Cumulative impacts of fishing pressure

Whereas the above one-year look-ahead elasticities address transient dynamics, it is also of interest how the system responds in the long-term to a scenario with constant fishing mortality rates. If the density dependence matrix $A$ permits a stationary system, then the elasticities of certain attributes of the stationary distribution may be investigated. In particular, it will be of interest to investigate how the stationary distribution responds to a proportional change in the unfished fraction of a functional group, $U_i = 1 - F_i$. From Eq. (2), note that $dU_i/(U_i\, df_i) = d\log U_i/df_i = -1$. Therefore a small perturbation of size $\Delta$ in the value of $f_i$ results in a $-\Delta$ proportional change in the unfished fraction $U_i$, which provides a useful interpretation of the fishing mortality rate.

Let $\tilde{x}(\theta, y) = \exp[-A^{-1}(r - f) + y(A, S)]$ denote attributes of the long-term stationary distribution of $x(t)$, where $y(A, S)$ is a function of the density dependent parameters $A$ and the process uncertainty matrix $S$. The vector $\tilde{x}(\theta, y)$ is equivalent to the long-term averages of the functional groups if $y(A, S) = \mathrm{diag}[P]/2$, and is equivalent to the long-term medians if $y(A, s) = 0$. In fact, due to the properties of the lognormal distribution (see, e.g., *Aitchison & Brown, 1957*, Ch. 2), the marginal $p$-quantiles of the stationary distribution of $x(t)$ may for any $p \in (0, 1)$ be obtained by choosing $y(A, S) = \mathrm{diag}[P]^{1/2} v_p$, where $v_p$ is the $p$-quantile of a standard normal distribution. The elasticities of $\tilde{x}(\theta, y)$ with respect to the unfished fractions are then given by,

$$\frac{\partial \log \tilde{x}(\theta, y)}{\partial \log U} = \frac{\partial \log \tilde{x}(\theta, y)}{\partial f} \frac{\partial f}{\partial \log U} = -A^{-1}, \tag{12}$$

where the proportional changes in $\tilde{x}(\theta, y)$ given perturbed fishing mortality rates are $\partial \log \tilde{x}(\theta, y)/\partial f = A^{-1}$ and $\partial f/\partial \log U = -I$. The elasticity matrix of Eq. (12) applies to the long-term means or the marginal $p$-quantiles, including the medians, of $x(t)$ presuming that the long-term distribution is stationary. The elasticity of functional group $i$ with respect to the unfished fraction of functional group $j$ is given by the $(i, j)$ entry of $-A^{-1}$. The elasticities of all functional groups with respect to the unfished biomass of the $j$th functional group are given by the $j$th column of the matrix $-A^{-1}$. This corresponds to a scenario where a fishery or set of fisheries (gear types) decrease aggregate fishing pressure on a particular functional group.

On the other hand, the scenario where the fishing pressure of a particular fishery or gear type is restricted may be of interest. In the Bay of Biscay, a particular gear type often targets more than one functional group (Fig. 1). Since elasticities can be added (*Caswell, 2001*, Ch. 9), the elasticities to simultaneous proportional increases of equal magnitudes in the unfished fractions of more than one functional groups are obtained by summing the columns of $-A^{-1}$ that correspond to the functional groups with decreased fishing pressure. Note also that an increase in fishing pressure corresponds to decreased unfished fractions, which simply reverses the sign of Eq. (12). Correspondingly, the response of a functional group to increased fishing pressure in one functional group and decreased fishing pressure

in another is given by the difference of the respective columns of $-A^{-1}$. Therefore both the direct and indirect long-term effects of fishing pressure on functional group biomasses due to changes in either single or multiple fisheries may be predicted from Eq. (12).

Four long-term management scenarios are explored, where the long-term stationary distribution is assessed for four choices of constant fishing mortality rates:

| | |
|---|---|
| $f^0$ | This scenario investigates the long-term stationary distribution given cessation of all fishing. |
| $f^{Avg}$ | This scenario sets the fishing mortality rates to the estimated average over the years 1999–2015. |
| $f^G$ | This scenario is as for $f^{Avg}$ except that the fishing mortality rates for pelagic piscivores $G$ is set to zero. This scenario examines the amount of compensation that might occur in the Bay of Biscay ecosystem from increased top-down control by pelagic piscivores released from fishing pressure. |
| $f^{MT}$ | This scenario is as for $f^{Avg}$ except that the fishing mortality rates are reduced proportional to the reduction in the average landings by French fisheries for ICES divisions 8a and 8b of a functional group from 1999 to 2015 with landings from mixed trawls excluded. The average proportion of benthivore landings attributed to mixed trawls was 62%, and the average proportion of demersal piscivore landings attributed to mixed trawls was 45%. The fishing mortality rates for benthivores $B$ and demersal piscivores $D$ are reduced to 38% and 55% of their $f^{Avg}$ values, respectively. The fishing mortality rates for pelagic planktivores and piscivores are as for $f^{Avg}$ because mixed trawls did not substantially contribute to landings for these latter two functional groups. |

## INFERENCE

### Priors

Table 1 summarises the prior information for the unknown parameters, concatenated into the vector of unknown parameters $\theta$, which includes the latent states $x(t = 1999)$, to form the prior density $p(\theta)$. A description of the information contained by each prior distribution is given below.

#### Fished fraction

Independent standard uniform priors placed equal probability density on all possible fished fractions of each functional group in each year. The fished fraction $F_i(t)$ for the $i$th functional group is related to the instantaneous fishing mortality rate $f_i(t)$ by Eq. (7).

#### Observation error and process uncertainty

If the observation error standard deviation is 0.03, then from Eq. (1) there is a 50/50 chance that the observed CPUE ($C_{ik}(t)/E_k(t)$) from fishery $k$ for functional group $i$ in year $t$ is roughly within 2% of $q_{ik}x_i(t)$. If the observation error standard deviation is increased an order of magnitude to 0.3 then there is a 50/50 chance that the observed CPUE from fishery $k$ for functional group $i$ in year $t$ is within ∼20% of $q_{ik}x_i(t)$. These

**Table 1 Priors for model parameters $\theta$.** Subscripts for functional groups are given by $i \in \{K, B, D, G\}$ and subscripts for the commercial fisheries and the survey are given by $k \in \{Dredge, Hook, Mixed\ trawl, Net, Pelagic\ trawl, Pot, Purse\ seine, Survey\}$. The distributions listed are lognormal with log-scale mean $m$ and log-scale variance $s^2$, $LN(m, s^2)$; uniform with bounds $a$ and $b$, $U(a, b)$; normal with mean $m$ and variance $s^2$, $N(m, s^2)$.

| Description | Parameter | Priors |
|---|---|---|
| Fished fraction | $F_i(t)$ | $U(0, 1)$ |
| Observation error standard deviation | $\sigma_k$ | $LN(-1.76, 1.13)$ |
| Process noise standard deviation | $\eta_i$ | $LN(-1.76, 1.13)$ |
| Initial latent states | $x_i(1999)$ | $LN(\log h_i(1998), 3.24)$ |
| Density dependence magnitude | $a_{ij}$ | $U(0, 1)$ |
| Density independent growth rate | $r_i|a_{ii}$ | $N(a_{ii}\log \bar{h}_i(1950:1998), a_{ii}^2)$ |
| Catchabilities | $q_{ik}$ | $LN(-\log \bar{h}_i(1950:1998), 11.7)$ |

levels of observation error roughly correspond to levels of small and large observation and process uncertainty variances respectively as suggested by a study of discrete time state space models with an exponential per capita growth function (*Staples, Taper & Dennis, 2004*). For the Bay of Biscay ecosystem, the lower value may be exceptionally low given the limitation of any particular fishery or survey to fully sample a functional group. Independent lognormal priors therefore specified only 0.05 probability that the observation error standard deviations were less than the lower value of 0.03, and specified 0.7 probability that the standard deviations were less than 0.30. The prior specified approximately 0.05 probability that the standard deviation is greater than 1; an observation error standard deviation of 1 gives a 50/50 chance that the observed CPUE from fishery $k$ for functional group $i$ in year $t$ is within a half-fold or two-fold change from $q_{ik}x_i(t)$. The log-normal prior is heavy-tailed and so accounted for the possibility of high observation error standard deviation for a gear type or fishery.

The process noise standard deviation describes the magnitude of the variation of the process disturbances $\epsilon_i(t)$ around the deterministic component of Eq. (8). An analogous interpretation as developed above for observation error applies to Eq. (8), where for example a process noise standard deviation of 0.3 provides a 50/50 chance that $x_i(t+1)$ is within ~20% of $\exp[\delta(t)]$. The process noise standard deviation priors were set identical to the observation error standard deviation to ease comparison.

### Initial latent states

The lognormal prior specified probability 0.2 to the event that the initial latent state $x_i(t = 1999)$ for the $i$th functional group was more than an order of magnitude greater or less than the reported landings in 1998, given by $h(t = 1998) = [56, 23, 35, 3]^\top$, for the $i$th functional group.

### Density dependence and density independence

The priors for the intra-functional group density dependence and density independence parameters of the $i$th functional group were developed jointly such that $p(r_i, a_{ii}) = p(r_i|a_{ii})p(a_{ii})$. The parameter $\kappa_i$ is defined as the non-zero equilibrium of the $i$th functional group in the absence of fishing, inter-functional group density dependence

and stochasticity. With these assumptions, $\kappa_i = \exp(r_i/a_{ii})$ from Eq. (5). The functional group biomasses are composed of many aggregated species and age cohorts for which unstable or oscillatory dynamics are judged unlikely. The prior ensures a monotonic approach to $\kappa_i$, which is a stable equilibrium point if $0 < a_{ii} < 2$ for each $i = K, B, D, G$. The sign of the derivative $dx_i(t+1)/dx_i(t)$ evaluated at $\kappa_i$ is determined by the sign of $1 - a_{ii}$. If $0 < a_{ii} < 1$ then this derivative is positive and the functional group biomass will approach $\kappa_i$ with a monotonic trajectory; if $1 < a_{ii} < 2$ then the derivative is negative and the functional group biomass will exhibit damped oscillations around $\kappa_i$. Oscillations around $\kappa_i$ were prohibited with a standard uniform prior placed on $a_{ii}$. For comparison of the intra-functional group density dependence parameters with $a_{ij}$, independent standard uniform priors were placed on the magnitudes of the inter-functional group parameters. The magnitude of density dependence was therefore equivalent a priori for all density dependent relationships.

The prior specification for $\kappa_i$ was based on independent information provided by the average landings from 1950 to 1998, denoted by $\bar{h}_i(1950:1998)$ for the $i$th functional group, where $\bar{h}(1950:1998) = [145, 30, 73, 27]^\top$ with units in kilotonnes for functional group $i = K, B, D, G$. A lognormal prior specified 0.5 probability that $\kappa_i$ is within a half-fold or two-fold change from the average landings $\bar{h}_i(1950:1998)$. Given the independent prior on the intra-functional group density dependence, the prior for the density independent growth rate was induced by the relationship $r_i = a_{ii}\log \kappa_i$ for a model without fishing, inter-functional group density dependence or stochasticity.

### Catchability

Conditional on the latent biomass $x_i(t)$ in the observation model given by Eq. (1), the catchability $q_{ik}$ defines the median catch per unit effort, $C_{ik}(t)/E_k(t)$. The catch per unit effort is arbitrarily defined depending on the choice of units for catch and effort. From the above lognormal prior for intra-functional group density dependence, the a priori median estimate of the carrying capacity is given by $\kappa_i$ for $i = K, B, D, G$. If the biomass of the $i$th functional group in year $t$ is equal to its carrying capacity so that $x_i(t) = \kappa_i$, then in the absence of observation error the observed CPUE is $C_{ik}(t)/E_k(t) = q_{ik}x_i(t) = 1$ when $q_{ik} = 1/\kappa_i$. The catchability prior was therefore centred at $1/\kappa_i$ for $i = K, B, D, G$, which would roughly correspond to the median value of effort having a similar magnitude as the median value of catch for a functional group with $\kappa_i$ biomass. The prior specified 0.50 probability that median catchability was within roughly an order of magnitude of $1/\kappa_i$, and 0.50 probability that catchability was outside of this range.

### Posterior density

The posterior distribution is defined by the process model of Eq. (9) for the latent states $x(t)$ in the years $t = 2000, \ldots, 2015$, by the observation model of Eq. (1) and by the prior distributions for the parameters, $\theta$, that include the unknown initial latent states, $x(t = 1999)$. The posterior density is given by,

$$p(x(1999:2000), \theta | C, E, h(1999:2015)) \propto p(\theta)p(h(1999) | x(1999), F(1999))$$

$$\times \prod_{t=2000}^{2015} p(x(t)|x(t-1),\theta) \left( \prod_i p(h_i(t)|x_i(t),F_i(t)) \left[ \prod_k p(C_{ik}(t)|x_i(t),E_k(t),\theta) \right] \right), \quad (13)$$

where $x(1999:2000)$ gives the latent biomasses of the functional groups for the years 1999–2000, $C$ is the array of annual catches by French commercial fisheries and scientific surveys that target specific functional groups for years 2000–2015, $E$ is the array of effort recorded by French commercial fisheries and scientific surveys for years 2000–2015, and $h(1999:2015)$ gives the recorded international landings for each functional group for years 1999–2015. The first line of Eq. (13) incorporates the priors for the parameters $\theta$ and the likelihood of the observed international landings in 1999 given the initial states. The second line of Eq. (13) incorporates the prior for the latent dynamic process of the unknown annual biomasses of the functional groups for years 2000–2015. The second line also includes the likelihood for the years 2000–2015 for harvest of functional groups $i \in \{K, B, D, G\}$ and gear type $k \in \{Dredge, Hook, Mixed\ trawl, Net, Pelagic\ trawl, Pot, Purse\ seine, Survey\}$. The $C_{ik}(t)$ are assumed non-negligible only for the targeted combinations of gear type and functional group shown in Fig. 1; catches for incidental combinations of gear type and functional group are assumed ignorable and do not contribute to the likelihood. The posterior distribution of a function $g(\theta)$ of the parameters $\theta$, such as the long-term stationary distribution of the latent states or the elasticities, is given by $p(g(\theta)|C, E, h)$.

## Estimation

Simulations and posterior inferences used R (*R Core Team, 2018*) with posterior samples obtained by Markov chain Monte Carlo (MCMC; see e.g., *Robert & Casella, 2004*; *Gelman et al., 2014*) using JAGS (*Plummer, 2003*) from R port rjags (*Plummer, 2016*). Five independent parallel chains were initialised by drawing realisations from the prior distributions except for the log catchabilities and the latent states. The log catchabilities were initialised to the average of the log CPUE indices. The latent state initialisations must not conflict with the censored observations, that is, the initial latent state trajectory must allow for a feasible harvest sequence. The latent states were therefore initialised from a Pareto distribution with cumulative distribution function given by $P(x_i(t = 1999) < w) = 1 - (h_i(t)/w)^2$ if $w \geq h_i(t)$ and 0 otherwise. A priori there is probability 0.75 that the fished fraction of functional group $i$ in year $t$ was less than 0.5. The Pareto prior is heavy-tailed and, with index parameter 2, has finite mean that is twice the observed value of landings and infinite variance.

The five independent parallel chains were each run for 500,000 adaptive iterations followed by 500,000 non-adaptive iterations with thinning rate of 10 applied to the non-adaptive phases. Trace plots were examined and Gelman and Rubin's convergence diagnostic (*Gelman & Rubin, 1992*) was applied using the coda package (*Plummer et al., 2006*). The upper bounds of the 95% confidence intervals for the estimated potential scale reduction factors were near 1 for every parameter. Results presented use the 250,000 posterior samples retained from the thinned non-adaptive phase. The code and data used for the estimation of the model are included as an R package (see Supplemental Information).

## MODEL ADEQUACY

The parameters of the Gompertz model have been shown to correspond to the one-year look-ahead elasticities of the unobserved functional group biomasses. To demonstrate the adequacy of the model for predictions, it would be desirable to compare biomasses predicted from the model to the true biomasses in the absence of observations for a given year. However, the functional group biomasses are never observed directly and so such a comparison is impossible. Instead, the best indicator of biomass in each year is given by the commercial and scientific survey catch per unit effort data. Therefore, predictive cross-validation was performed on the CPUE data in each year of the observation period.

*Gelfand (1996)* defines the cross-validation predictive density as the quantity

$$p(y_j | y_{1:n\setminus j}) = \int p\big(y_j | y_{1:n\setminus j}, \psi\big) p\big(\psi | y_{1:n\setminus j}\big),$$

where $y_j$ is the $j$th of $n$ observations and $y_{1:n\setminus j}$ is the set of $n-1$ observations excluding $y_j$. The unknown $\psi$ are unobserved quantities, such as static parameters or latent states. In the terminology of cross-validation, $y_j$ is the hold-out datum and $y_{1:n\setminus j}$ is the training data. If samples from the set of cross-validation predictive densities $\{p(y_j | y_{1:n\setminus j}) : j = 1, \ldots, n\}$ may be computed by MCMC, then *Gelfand (1996)* suggests the following measure of model adequacy. Samples from each cross-validation predictive density $p(y_j | y_{1:n\setminus j})$ may be obtained, which can then be used to construct central cross-validation predictive intervals of probability $\alpha$ for each $j$. The proportion of the $n$ observations that fall within their corresponding central cross-validation predictive intervals should then be close to $\alpha$.

Rather than leaving out only a single observation, instead a block of more than one observation may be held out at the same time (e.g., *Geisser, 1975*). The set of predictive densities of interest is then $\{p(y_j | y_{1:j\setminus v(j)}) : j = 1, \ldots, n\}$, where $v(j)$ indicates the $v$th block of observations that includes $y_j$. The $n$ observations may be allocated to the blocks randomly or instead in a way that reflects the dependence structure of the model. As for the leave-one-out cross-validation test of model adequacy described above, the proportion of the $n$ observations that fall within their corresponding central cross-validation predictive intervals should then be close to $\alpha$.

The test for model adequacy in this study focuses on the ability of the model to predict CPUE for all functional groups and all gear types in a given year. The interest therefore focuses on the following set of cross-validation predictive densities. Let $D_t = \{C_{ik}(t)/E_k(t) : \forall i, k\}$ denote the set of all CPUE data in year $t$, which includes missing values for non-targeted combinations of gear type and functional groups (see Fig. 1). Let $D_{t_1:t_3}$ denote the collection of all CPUE data from year $t_1$ to year $t_3$, and let $D_{t_1:t_3\setminus t_2}$ denote the collection of all CPUE data from year $t_1$ to year $t_3$ exclusive of CPUE data from year $t_2$. For a given year $t$, $D_t$ is the hold-out data and $D_{2000:2015\setminus t}$ is the training data. The joint cross-validation predictive density of all CPUE data for each year $t \in \{2000, \ldots, 2015\}$ is given by,

$$p\big(D_t | D_{2000:2015\setminus t}, h(1999:2015)\big) = \int p(D_t | x(t), \theta) p\big(x(t), \theta | D_{2000:2015\setminus t}, h(1999:2015)\big)$$
$$\times dx(t) d\theta.$$

Samples from the marginal posterior distribution of the static parameters and the latent functional group biomasses in year $t$ were obtained conditional on $D_{2000:2015\backslash t}$. For a given year $t$, the CPUE data from that year were removed and estimation otherwise proceeded as described in the *Inference–Estimation* section above. Conditional on each joint sample of the static parameters and latent biomasses, a sample of CPUE was jointly simulated from the observation models of Eq. (1) for each combination of gear type and functional group. From these samples, quantiles and central credible intervals were computed for each of the cross-validation predictive densities in the set $\{p(C_{ik}(t)/E_k(t)|D_{2000:2015\backslash t}, h(1999:2015)) : t = 2000, \ldots, 2015, \; \forall i, k\}$, that is, all combinations of functional groups and gear types in each year of the 16 year observation period. The cross-validation predictive density is empty for a non-targeted combination of gear type and functional group with no corresponding observations. The proportions of the 288 observations from targeted combinations of gear types and functional groups that were below the median, within the 50% and within the 90% central intervals obtained from the corresponding predictive posterior distributions were computed. These proportions should be expected to be close to 50%, 50% and 90%, respectively.

## RESULTS

### Observation model

The observation model uncertainty provided intuitively reasonable results (Fig. 2). As noted in the prior description, an observation standard deviation of 1 corresponds to a 50/50 chance that the observed CPUE in year $t$ is within one-half or $2\times$ the true biomass. A standard deviation of 0.3 gives 0.5 probability that the CPUE is within $\pm20\%$, and a standard deviation of 0.03 give 0.5 probability of being within a $\pm2\%$ margin. The observation error standard deviation estimates were highest (had greatest uncertainty) for purse seines and were lowest for mixed trawls (Fig. 2). Thus the trawl fishery yielded greater precision than the purse seiners. The related parameter of catchability was lowest when looking at rare functional group and gear combinations. For example, estimated catchability was higher for both benthivores and demersal piscivores than pelagic piscivores in the netter fishery (Table S2). On the other hand, catchability for pelagic planktivores was higher than demersal piscivores in pelagic trawls. Estimated catchability was higher for planktivores then pelagic piscivores in the purse seine fishery.

### Ecosystem dynamics

There are two main conclusions for latent biomass states (Fig. 3). First, there were common qualitative trends evident for benthivores and both demersal and pelagic piscivores. These functional groups appear to have declined from 2000 to the late 2000s before increasing for the remaining duration of the observation period. This same pattern is shown for the total biomass (Fig. S3). The pattern of increase and decrease in the one-year look-ahead elasticities for fishing (Fig. 4) was similar. Figure 4 shows annual fishing elasticities were increasingly negative for all functional groups into the late 2000s, followed by an easing toward zero, except for planktivores, where the estimated fishing elasticities became increasingly negative again after 2011. Unlike the other functional groups, planktivores

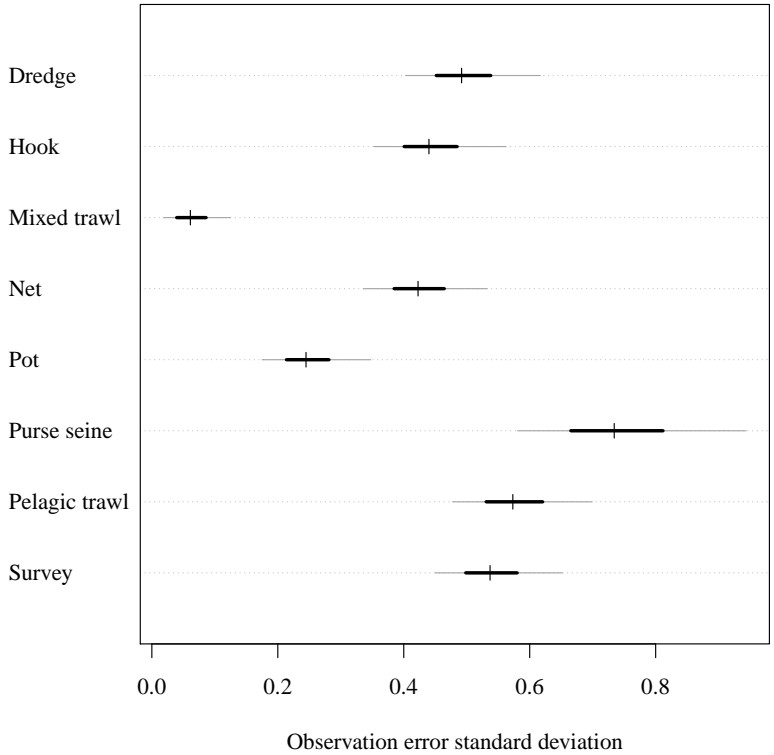

**Figure 2  Estimated observation error standard deviation ($\sigma_i$) for commercial fishery gear types and fishery independent scientific survey.** Posterior median given by hashes, 50% central credible intervals by black segments and 90% central CIs by grey segments.

appeared to plateau and perhaps slightly decrease toward the end of the the observation period (Fig. 3).

The one-year look-ahead elasticities for fishing were closely related to the amount of biomass in the Bay of Biscay ecosystem. The median elasticities of the fishing mortality rates were of the greatest magnitude for all functional groups during the years 2006 or 2007 (Fig. 4). These years corresponded to a period of low biomass (Fig. 3) and high fished fractions (Fig. S4). For planktivores, the magnitude of the fishing mortality elasticity was also high in 2015. This year corresponded to a slight decrease in the estimated biomass of planktivores and an elevated fished fraction relative to the preceding years. In contrast, the fishing mortality elasticity was at a low magnitude for pelagic piscivores in 2015. Pelagic piscivores had at this time an estimated upward trend in biomass and a relatively low fished fraction compared to preceding years.

The density dependence parameters are directly interpretable as elasticities of the one-year look-ahead predictions with respect to functional group biomasses. The posterior mean elasticities for the density dependence relationships are shown in Fig. 5. On average, intra-functional group density dependent elasticities were greater for the pelagic functional groups than for the demersal functional groups. Uncertainty in the estimated intra-functional group density dependent elasticities was however high for both pelagic

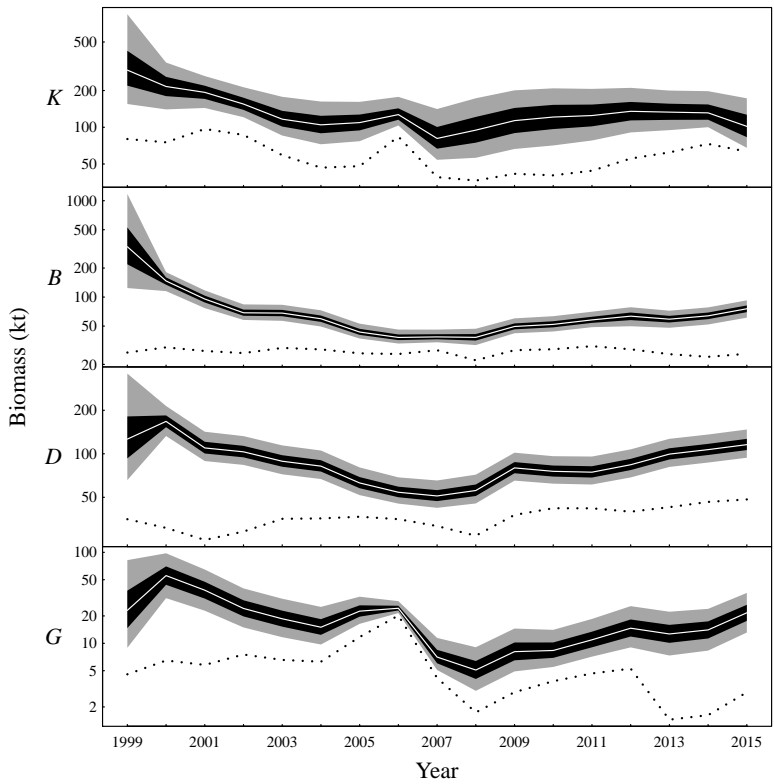

**Figure 3** **Latent state estimates of fishable biomass for the functional groups (*K*: planktivores, *B*: benthivores, *D*: demersal piscivores, *G*: pelagic piscivores).** The dotted line is the recorded annual harvest for each functional group. Posterior medians given by white lines, 50% central credible intervals by black polygons and 90% central CIs by grey polygons. Note logarithmic *y*-axes.

functional groups (Fig. S5), and comparatively low for demersal functional groups. For pelagic functional groups, it is possible that migratory behaviour increased the variability of intra-functional group density dependence.

Turning to inter-functional group elasticities, Fig. 5 shows that the density-dependent elasticities were higher between pelagic piscivores and benthivores than between pelagic piscivores and planktivores. The positive effect of planktivores on the consumer functional groups on average outweighed the negative predation effect of consumers on planktivores (Fig. 5). The mean estimates shown in Fig. 5 were however accompanied by uncertainty (Fig. S5) with higher magnitudes of inter-functional group density dependence apparently more likely for bottom-up rather than top-down effects. The bottom-up versus top-down impact was therefore compared for each density relationship between the consumer and resource functional groups. The density dependent influence of planktivores on its consumers was somewhat more likely to be stronger than top-down predation pressure for both demersal ($P(a_{DK} > a_{KD}|C, E, h) = 0.73$) and pelagic piscivore ($P(a_{GK} > a_{KG}|C, E, h) = 0.73$) trophic pathways. However, the comparison between top-down and bottom-up magnitudes was more equivocal for benthivores. The posterior probability that the density dependent influence of benthivores on demersal piscivores was greater than the predation pressure

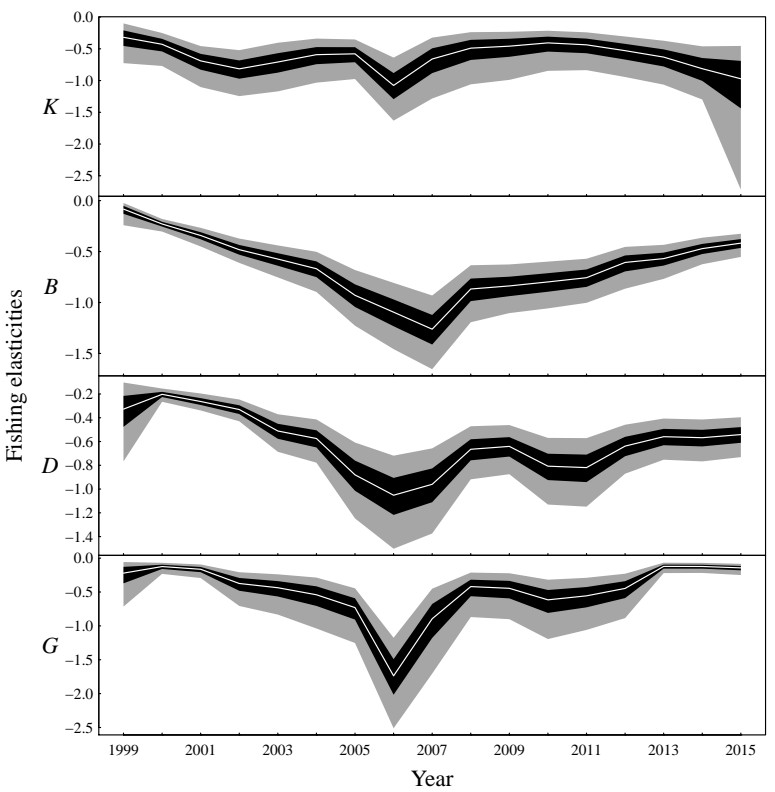

**Figure 4** Elasticities of the one-year look-ahead predictions to changes in the fishing mortality rate of the fishable biomass for the functional groups (**K**: planktivores, **B**: benthivores, **D**: demersal piscivores, **G**: pelagic piscivores). Posterior medians are given by white lines, 50% central credible intervals by black polygons and 90% central CIs by grey polygons.

was only $P(a_{DB} > a_{BD}|C, E, h) = 0.58$. The posterior probability of a stronger bottom-up benthivore effect on pelagic piscivores was equally likely to a stronger top-down effect ($P(a_{GB} > a_{BG}|C, E, h) = 0.49$).

A further analysis examined the probability that the elasticity magnitudes for inter-functional group density dependence exceeded 0.2 (Table 2). A density dependent elasticity with a magnitude of $a_{ij} = 0.2$ predicts that increasing functional group $j$ by a proportional amount would change the biomass of functional group $i$ in the next year by 20% of the same proportional amount. For each pairwise trophic relationship, the bottom-up effect of a resource (prey) functional group on its predator had a greater chance of exceeding 0.2 than the magnitude of the top-down predation effect. This result shows that high magnitudes of density dependence were more likely for positive bottom up effects of prey functional groups ($K$ and $B$) on their predators ($D$ and $G$) than for negative predation effects of the consumer functional groups on the prey functional groups. The greater magnitude of bottom-up versus top-down density dependence was however more pronounced for planktivores than for benthivores (Table 2). Whereas the probabilities that the bottom-up elasticities exceeded the 0.2 threshold were an order of magnitude greater than the top-down
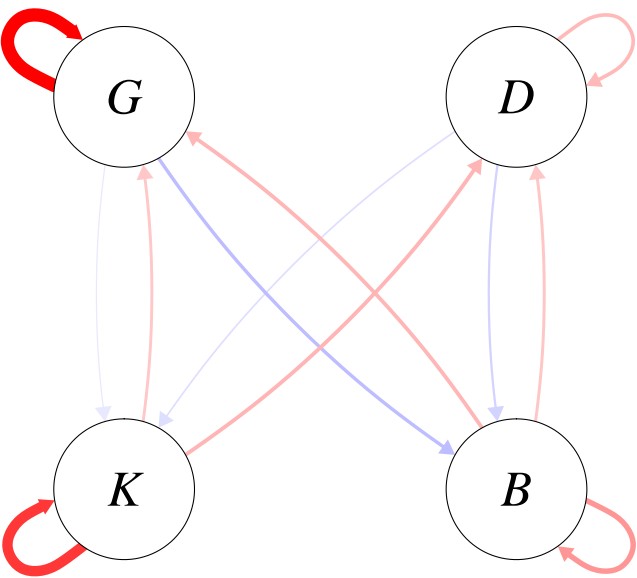

**Figure 5** **Elasticities of one-year look-ahead predictions of functional groups to changes in the previous year's biomass for the functional groups (*K*: planktivores, *B*: benthivores, *D*: demersal piscivores, *G*: pelagic piscivores).** Plotted are posterior means; positive elasticities are in red and negative elasticities are in blue; width of arrows is relative to the magnitude of the posterior mean elasticity.

**Table 2** **Threshold exceedance probabilities for elasticities of the one-year look-ahead predictions with respect to inter-functional group density dependence.** The posterior probabilities that the magnitudes of elasticities exceed 0.2 are given for the bottom-up effects of resource functional groups (*K* and *B*) on consumer functional groups (*D* and *G*) in the first column, and the top-down effects of consumer on resource functional groups in the second column.

| | $P\left(a_{ij} > 0.2 \mid C, E, h(1999:2015)\right)$ | |
| --- | --- | --- |
| | **Bottom-up** | **Top-down** |
| $K - D$ | 0.20 | 0.02 |
| $K - G$ | 0.10 | 0.01 |
| $B - D$ | 0.11 | 0.05 |
| $B - G$ | 0.19 | 0.10 |

elasticities for planktivores, for benthivores the exceedance probabilities for bottom-up elasticities were only about double that of the top-down elasticities.

Density independent elasticities were lowest for the pelagic piscivores compared to the other functional groups (Fig. S6). This result may reflect the greater magnitude of process error standard deviation estimated for pelagic piscivores compared to the other functional groups (Fig. S7). High process uncertainty may lead to dynamics driven relatively more by stochasticity rather than the deterministic dynamics of Eq. (5). As noted above, uncertainty was also high for estimates of intra-functional group density dependence of pelagic piscivores, which together with high process uncertainty may reflect an increased importance of migration for this functional group.

## Cumulative impacts of fishing pressure

The density dependence captured by the matrix $A$ determines whether or not the ecosystem model is stationary. For stationarity, the maximum of the moduli of the eigenvalues of the transition matrix $I + A$ must be less than one. The Bay of Biscay functional group ecosystem model was stationary with posterior probability of 1. Relative to the unfished long-term distribution of biomass (scenario $f^0$ in Fig. 6), the scenario of average fishing mortality $f^{Avg}$ reduced the stationary distributions of biomass for all functional groups (Fig. 6).

The estimated posterior median elasticities of the long-term averages, medians or other arbitrary quantiles of the stationary distribution (denoted by $\tilde{x}(\theta, y)$, see Eq. (12)) with respect to the unfished fractions $U$ were

$$\frac{\partial \log \tilde{x}(\theta, y)}{\partial \log U} = -A^{-1} = \begin{array}{c} K \\ B \\ D \\ G \end{array} \begin{pmatrix} K & B & D & G \\ 1.46 & -0.02 & -0.06 & -0.07 \\ -0.04 & 1.15 & -0.07 & -0.20 \\ 0.16 & 0.10 & 1.07 & -0.03 \\ 0.19 & 0.20 & -0.02 & 1.68 \end{pmatrix}.$$

Table S3 provides the posterior interquartile distances as a measure of the uncertainty associated with the above estimates of the long-term elasticities. The scenario $f^G$ that removes all fishing of pelagic piscivores corresponds to increasing the unfished fraction of pelagic piscivores, and so the last column of the above matrix applies. These elasticities predict that increasing the unfished fraction of pelagic piscivores (equivalently, decreasing the fishing mortality rate $f^G$) substantially increases pelagic piscivore fishable biomass and also slightly decreases benthivore fishable biomass, in the long-term, relative to scenario $f^{Avg}$. Figure 6 shows that the biomass of pelagic piscivores increased relative to the scenario $f^{Avg}$ of average fishing mortality. Also, the median biomass of benthivores was slightly lowered in scenario $f^G$ because of the greater biomass of predators retained in the ecosystem. The landings of benthivores were also reduced relative to the average scenario $f^{Avg}$ (Fig. S8). The biomass and landings of planktivores and demersal piscivores were relatively less affected.

From Fig. 1, the mixed trawl fishery targets both benthivores and demersal piscivores. Eliminating effort from the mixed trawl fishery (scenario $f^{MT}$) corresponds to a reduction of the fishing mortality rate to 38% of its $f^{Avg}$ value for benthivores and to 55% of its $f^{Avg}$ value for demersal piscivores. The unfished fraction is thereby increased for both of these functional groups, and so the cumulative elasticities are obtained by summing the two middle columns of the above matrix $-A^{-1}$. The resulting predictions are of substantial increases for both benthivores and demersal piscivores, and also a slight increase for pelagic piscivores for the scenario $f^{MT}$ relative to $f^{Avg}$. Figure 6 shows that the biomass of benthivores increased to roughly the same level as in the unfished scenario ($f^0$), and the biomass of demersal piscivores also increased. The relative difference between the posterior medians of the long-term biomass of pelagic piscivores also increased by $\sim$9% when moving from scenario $f^{Avg}$ to $f^{MT}$. The landings of benthivores had the greatest reduction in this last scenario (Fig. S8).

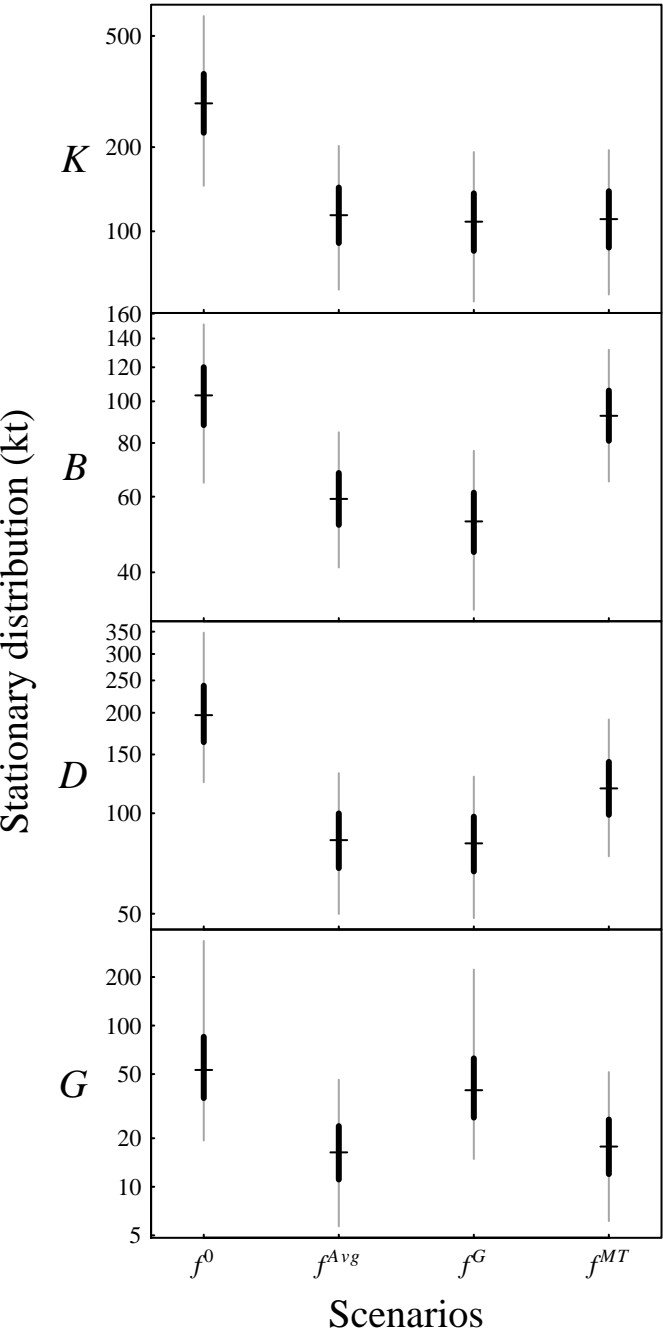

**Figure 6** Plotted are the posterior stationary distributions under the four management scenarios for the functional groups (*K*: planktivores, *B*: benthivores, *D*: demersal piscivores, *G*: pelagic piscivores). The central hashes indicate the posterior median, the black segments indicate the central 50% CIs and the grey segments indicate the central 90% CIs.

## Predictive cross-validation and model adequacy

Predictive cross-validation of the CPUE data was performed across all possible single-year hold-outs from the 2000 to 2015 observation period. Accounting for all targeted gear type and functional group combinations yielded a total of 288 CPUE observations and predictions across the 16 years (Fig. 7). About one-half of the observations were below the corresponding cross-validation predictive medians. About 56% of the observations were within the corresponding 50% central cross-validation predictive intervals, and about 90% of the observations were within the corresponding 90% central cross-validation predictive intervals.

# DISCUSSION

## Modelling

The Gompertz model parameters were shown to be directly interpretable in terms of the one-year look-ahead prediction elasticities, which measure the magnitude of the proportional change in next year's biomass given a proportional change to a parameter or variable in the current year. In particular, these elasticities provided an intuitive interpretation of the time-invariant density dependent parameters within the model, where from Eq. (10) each entry $A_{ij}$ in the matrix $A$ is the elasticity of functional group $i$ given a small perturbation to the biomass of the $j$th functional group. The negative of the inverse of the density dependent matrix, $-A^{-1}$ is on the other hand shown to determine the long-term elasticities of functional group biomasses with respect to their unfished fractions. These long-term elasticities apply to either the long-term means or any choice of marginal quantiles from the stationary distribution.

An advantage of the modelling approach is that it provides coherent estimates of model parameters and functional group biomasses as well as ecosystem process and observation errors. The analysis allowed joint estimation of unknown fishing, process and observation model parameters in addition to the unobserved latent states of functional group biomasses. The joint estimation of these unknown factors is in contrast to most food web models currently used. The price to pay for this coherence is that model complexity needs to be reasonable. The Gompertz model used in this paper, which is a linear process model with multivariate normal process and observation error on the logarithmic scale, is a common choice for the analysis of multivariate ecological time series (e.g., *Ives et al., 2003*; *Spencer & Ianelli, 2005*; *Lindegren et al., 2009*; *Bell, Fogarty & Collie, 2014*; *Torres et al., 2017*). Depending on how fishing pressure is modelled, the log-linear normal model may allow for application of analytical updating schemes for the latent states (e.g., *Thompson, 1996*). For Bayesian state space ecological models with unknown magnitudes of observation error and process noise, alternative choices may for example accommodate non-Gaussian observation error (e.g., *Hosack, Peters & Hayes, 2012*) or non-linear and non-Gaussian process models (e.g., *Hosack, Peters & Ludsin, 2014*). On the other hand, the Gompertz model, as a log-linear normal model for the latent states, is a reasonable assumption for many ecological populations (*Dennis et al., 2006*). The stationary distribution of the Gompertz model is also easily accessible, which accommodates long-term predictions of ecosystem response to management scenarios while accounting for uncertainty.

Cross−Validation Predictive Densities of CPUE by Year

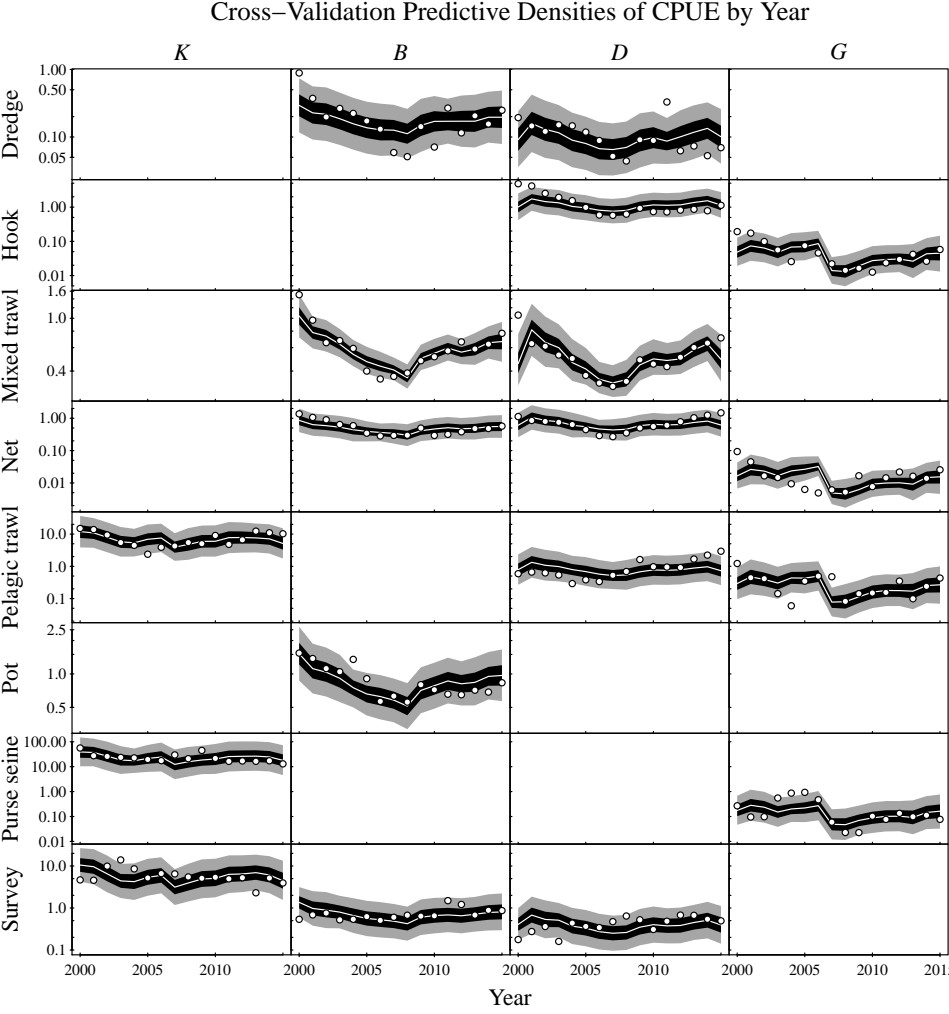

**Figure 7** **Cross-validation predictive densities blocked by year for the observation period that extends from 2000 to 2015.** For each hold-out year, the cross-validation predictive medians for the CPUE conditioned on the observed CPUE from all remaining years are given by white lines, 50% central intervals by black polygons and 90% central intervals by grey polygons. The observed CPUE data are plotted as points. Functional groups ($K$: planktivores, $B$: benthivores, $D$: demersal piscivores, and $G$: pelagic piscivores) are ordered by column and gear types by row. Non-targeted combinations of gear type and functional groups are empty. CPUE units are in metric tonnes per day of fishing for commercial fisheries and metric tonnes per km$^2$ for the scientific surveys. Note logarithmic $y$-axes.

As an assessment of model adequacy, the Gompertz model was shown to have predictive capability by comparing the cross-validation predictive densities for hold-out years against observed CPUE (Fig. 7). The proportions of observations below the cross-validation predictive medians of the hold-out years was about 50%, as expected. Similarly, the proportions of observations within the 50% and 90% central cross-validation predictive intervals were close to 50% and 90% respectively, even though sometimes conflicting information was contributed by the various commercial and scientific survey CPUE time series. For example, the purse seine CPUE of pelagic piscivores ($G$) decreased moving into
the last year of the time series (2015), whereas the observed CPUE increased for all other targeted gear types of pelagic piscivores in that year. Nevertheless, the cross-validation predictive densities for the hold-out years achieved by the model appeared reasonable (Fig. 7). The demonstrated predictive capability of the multivariate Gompertz model increases confidence in the usefulness of the interpretable one-year look-ahead elasticities and the importance of jointly estimating unknown process and observation uncertainty in the state space model framework.

It should be noted that alternative discretisation schemes are commonly applied to continuous-time theoretical models originally formulated as an ordinary differential equation (ODE) when deriving discrete-time models (e.g., *Turchin, 2003*), such as that used in this paper. In fisheries, a common form of discretisation has the structure

$$x_i(t+1) = x_i(t) + G(x_i(t)) - C(t),$$

where $C(t)$ is the harvest or catch in year $t$. The choice of the function $G(\cdot)$ captures different forms of density dependent growth (e.g., *Hilborn & Walters, 1992*; *Haddon, 2011*). The function $G(\cdot)$ is usually inspired by the form of density dependence as represented by continuous-time models such as the Schaefer model (*Schaefer, 1954*), Pella-Tomlinson model (*Pella & Tomlinson, 1969*) or Gompertz-Fox model (*Fox, 1970*). For the form of discretisation above, high levels of harvest in year $t$ can lead to the prediction that next year's stock size is negative, which is a biological impossibility. The discretised form of the Gompertz model used in the paper avoids this problem by instead choosing the exponential per capita growth function of Eq. (8), which precludes biologically impossible predictions of negative stock size. The resulting equilibrium of the deterministic version of the discrete multivariate Gompertz model used here is shown to be identical to the multivariate generalisation of the original deterministic continuous-time Gompertz-Fox ODE model introduced by *Fox (1970)*.

## Ecosystem dynamics

After a decrease in the early 2000s, model results estimate that total (fishable) biomass steadily increased in the Bay of Biscay ecosytem from around 2008. This pattern was supported by increases in all functional groups, except planktivores, which remained stable during the latter portion of the observation period. The model results reflect the decreasing trends that are evident for several of the CPUE time series early in the study period (Fig. S2). In the Bay of Biscay during the early 1990s, ten out of 20 assessed fish stocks were considered overexploited and 28 fish species were classed as subject to concern by the IUCN (*Lorance et al., 2009*). Further, empirical analysis indicates that total demersal biomass had remained stable between the late 1980s and the beginning of the study period (*Rochet et al., 2005*). Thus, the Bay of Biscay ecosystem was clearly impacted by overfishing prior to the study period and seemed to have remained so during the first part of the study period, despite the fact that fishing power has decreased since the early 1990s (*Mesnil, 2008*) with an acceleration in the early 2000s (*Rochet, Daurès & Trenkel, 2012*). However, time delays and functional groups responding in different time frames to a reduction in fishing pressure is expected due to differences in life history traits (*Collie, Rochet & Bell, 2013*). Planktivores

are expected to rebound faster, which could explain the stability of biomass found here for this group.

A previous study from the neighbouring Celtic Sea found similar patterns of biomass fluctuations for a short period of overlapping years in the early 2000s (*Bell, Fogarty & Collie, 2014*). That study analysed the same functional groups in the years 1987–2004 while also using a Gompertz model, although the analysis differed in other respects such as the way in which fishing pressure was included and the inferential framework. Nevertheless, *Bell, Fogarty & Collie (2014)* estimated decreasing total biomass in the Celtic Sea during the early 2000s. All functional groups were estimated to be decreasing in the Bay of Biscay during this period (Fig. 3 and Fig. S3). The Celtic Sea study did suggest an increase in planktivores during the early 2000s. However, only trawl survey data and no commerical CPUE was included (*Bell, Fogarty & Collie, 2014*). The survey CPUE from the Bay of Biscay also suggest a small increase in planktivore biomass in the early 2000s, whereas the CPUE from nets and purse seines suggest a decrease (Fig. S2). The estimated latent biomass of planktivores (Fig. 3) appears to track these latter CPUE indices rather than the scientific survey data during this period in the Bay of Biscay. The scientific surveys, although fishery-independent, also have a limited spatio-temporal footprint compared to the commercial fisheries. The scientific survey may also target different age classes, size classes or species from fisheries that target fishable biomass. The commercial fishery and scientific surveys were therefore a priori given equal weightings by placing independent and identical priors on the observation error standard deviations associated with each gear type. The relative weighting of the scientific survey, or any of the commercial fisheries, could however be altered by changing these priors to capture additional information contributed by expert opinion or empirical data.

Significant, though variable, intra-functional group and inter-functional group density dependence was identified in the Bay of Biscay. The estimated intra-functional group density dependence was more uncertain for the pelagic functional groups than the demersal functional groups. This difference between demersal and pelagic groups could be a result of density mediated food and habitat limitations being more important for the demersal groups, whereas environmental conditions are known to be important drivers of population dynamics for pelagic species, in particular small and medium sized pelagics (*Trenkel et al., 2014*). The predator functional group of pelagic piscivores also exhibited high uncertainty in process model uncertainty (e.g., the process noise standard deviation). For pelagic piscivores, it is possible that migratory behaviour increases the uncertainty of intra-functional group density dependence, where year to year changes in biomass may be more sensitive to allochthonous import and export rather than endogenous production.

Top-down control by predators is expected to dampen ecosystem level fluctuations in marine ecosystems (*Cury, Shannon & Shin, 2003*). Although some food-web models are pre-determined as either bottom-up or top-down by construction e.g., *Steele (2009)*, the ability to estimate the strength of top-down and bottom-up relationships for each predator–prey pair is an advantage of the state-space modelling approach. There was evidence from the one-year look-ahead elasticities for top-down control being more important for benthivores in the Bay of Biscay than top-down control of planktivores

(Fig. 5 and Fig. S5). Overall, however, high magnitudes of elasticities were more probable for bottom-up effects of producers on consumers than top-down effects (Table 2). Using a detailed simulation model, *Oken & Essington (2015)* found that a top-down predator effect can be difficult to detect in a biomass production model, in particular if recruitment variation and observation error are high, or a second predator also targets the species. All of these factors occur in the Bay of Biscay ecosystem, which may have made detection of top-down control difficult.

## Fishing

Seven fishing fleets were defined by broad gear categories that exploited from one functional group (potters) to three functional groups (pelagic trawlers and netters). To make catchability estimates as much as possible comparable among functional groups within fleets, but also among fleets, the fishing effort was defined by the common currency of days-fishing. The fleet with the highest catchability was purse seine fishing on planktivores (posterior median catchability equal to 0.18, Table S2). This result is not surprising given the type of fishing method and the relatively small size range of the species making up the group. The lowest catchability was estimated for netters fishing pelagic piscivores (median catchability equal to 0.001). Interestingly, the catchabilities of demersal piscivores was comparable for mixed trawlers (median catchability 0.006) and the bottom trawl used in the fishery independent survey (median catchability 0.005). A higher catchability might have been expected for mixed trawlers given that this fishery targets large individuals above the minimum landing size. However, the observation error standard deviation was lowest for mixed trawls compared to all other gear types including the scientific survey.

The model results presented are of fishable biomass exploitable by the commercial fisheries that operate in the Bay of Biscay. Estimates of total biomass would require independent estimates of functional group catchabilities for the commercial fisheries. To the best of our knowledge, this is the first time catchability coefficients have been estimated on the functional group level within a state space model. Typically, catchability is estimated at the species level either experimentally (e.g., *Myers & Hoenig, 1997*; *Millar & Fryer, 1999*) or at the population level in stock assessment models. Time-varying catchability may also be estimated within a state-space model framework (*Wilberg et al., 2009*). For example, *Hosack, Peters & Ludsin (2014)* found evidence that catchability of commercial fisheries may vary with a proxy for habitat quality in a joint two-species state space model with unknown observation error and process uncertainty. However, the difficulties of parameterising complex models of catchability are heightened for ecosystem models that take into account many species with alternative life history strategies and behaviour. For the scientific survey used in this study, catchability has been estimated experimentally revealing large variability among species (*Trenkel & Skaug, 2005*). Such species level differences are averaged out at the functional group level estimates obtained here. Indeed, functional group and community level survey estimates have been found more robust to variability (season or gear type) compared to the species level (*Trenkel et al., 2004*).

The time-varying elasticities for fishing mortality showed how small perturbations of fishing mortality rates have temporally varying strong or weak effects on the one-year

look-ahead predictions (Fig. 4). When biomass was low and the fished fraction of a functional group was high, then the changes in the fishing mortality rate had stronger effects on the one-year look-ahead predictions compared to periods when biomass was high and the fished fraction low. At first it may seem surprising that pelagic piscivores supported comparably high rates of fishing mortality as other functional groups (Fig. 4) despite having the lowest rates of density independent growth ($r_G$ in Fig. S6). However, pelagic piscivores received positive biomass inputs from the density dependent trophic relationships with planktivores and benthivores (Fig. 5 and Fig. S5) that counteracted the low rates of density independent growth. The implication is that the level of sustainable fishing pressure for a given functional group may depend on other functional groups in the ecosystem.

Assessing the cumulative effects of multiple fisheries on marine ecosystems continues to be a concern for fisheries management (*Hobday et al., 2011*; *Link & Marshak, 2019*; *Zhou et al., in press*). However, the complex and nonlinear dynamics of marine ecosystems limit their predictability (*Glaser et al., 2014*; *Planque, 2016*). Therefore the long-term predictions of only four rather extreme fishing scenarios were studied. The results nevertheless provide insight into the potential effects of interactions between fisheries management and food web dynamics caused by direct and indirect effects. The posterior stationary distribution under each scenario conformed with predictions derived from the long-term elasticities of the biomasses with respect to changes in fishing pressure (see Eq. (12)). The scenario with no fishing on pelagic piscivores ($f^G$), which might be expected to increase top-down control on planktivores, led to a similar stationary biomass distribution of planktivores as status quo fishing ($f^{Avg}$) or excluding mixed trawlers ($f^{MT}$). The scenario $f^G$ instead decreased the stationary biomass of benthivores: the inter-functional group density dependence relationship with the greatest (median or mean) magnitude was the negative density dependent effect of pelagic piscivores on benthivores (Fig. 5 and Fig. S5). Both of these results agreed with the long-term elasticities of the biomasses with respect to changes in fishing pressure. Not surprisingly, the scenario with no mixed trawlers ($f^{MT}$) mainly benefited benthivores and demersal piscivores relative to the status quo fishing scenario ($f^{Avg}$) due to the release of fishing pressure on these two functional groups, although the biomass of pelagic piscivores also increased slightly. Again these results agreed with predictions from the long-term elasticities. From a fisheries management point of view the scenario results indicate that indirect effects might be expected in the Bay of Biscay when taking strong management actions for certain fleets or functional groups.

## CONCLUSIONS

The fishable biomass of functional groups in the Bay of Biscay was estimated within a state space model framework. The biomass and parameter estimates account for uncertainty in the process model dynamics, catchabilities and observation error associated with the various gear types that operate in the Bay of Biscay. The multivariate discrete-time Gompertz model derived from the continuous-time Fox model was shown to have interpretable parameters that give both transitory and long-term elasticities. The transitory elasticities correspond

to one-year look-ahead elasticities given a perturbation to the density independent rate of growth, the biomass of a functional group, or the instantaneous rate of fishing mortality. Predictive cross-validation showed that the state space model performed well in predicting functional group changes at the annual time scale. The elasticity of the long-term means or quantiles of the functional groups to sustained changes in fishing pressure were shown to be determined by the density-dependent parameters. Indirect effects among functional groups were predicted to occur for certain management actions if sustained over long time periods.

## ACKNOWLEDGEMENTS

We thank the Direction des pêches maritimes et de l'aquaculture for providing the French effort and landings data and are grateful to all collegues who participated in the survey. We thank Nicholas Beeton, the editor and two anonymous reviewers for their helpful comments and suggestions. The design, analysis and results of the study are the sole responsibility of the authors and did not engage the data providers or funders.

### Funding

This work was supported by the Eranet Cofasp project PrimeTradeOffs (ANR-15-COFA-0004-01). Additionally, the European Data Collection Framework co-funded the Evhoe scientific survey. The funder had no role in study design, data collection and analysis, decision to publish, or preparation of the manuscript.

### Grant Disclosures

The following grant information was disclosed by the authors:
Eranet Cofasp project PrimeTradeOffs: ANR-15-COFA-0004-01.
Evhoe scientific survey.

### Competing Interests

The authors declare there are no competing interests.

### Author Contributions

- Geoffrey R. Hosack and Verena M. Trenkel conceived and designed the experiments, performed the experiments, analyzed the data, contributed reagents/materials/analysis tools, prepared figures and/or tables, authored or reviewed drafts of the paper, approved the final draft.

### Data Availability

   The model code and data are available in the Supplemental Files as an R package.

### Supplemental Information

Supplemental information for this article can be found online at http://dx.doi.org/10.7717/peerj.7422#supplemental-information.

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
