# Peer review of "Functional group based marine ecosystem assessment for the Bay of Biscay via elasticity analysis"

_PeerJ, doi:10.7717/peerj.7422_

## Round 0.1 · original submission · Major Revisions

I enjoyed reading the paper - as one reviewer pointed out, it is hard reading for ecologists who have not done this kind of modelling before, but it is at the same time not easy to explain in a few words what the different equations mean in terms of biological questions and assumptions. This reviewer suggested ways to improve the manuscript, and I let you decide how you can make the necessary changes.
I had a few additional comments:

1) You do not show the predictions from the model against the observed values (that is not the fitted values as shown in different figures, but using the model to predict next year observations; you can discuss how some terms need to be included to calculate predictions). One is left with elasticities as being an assessment of the model, but no idea regarding how good these elasticities will represent actual changes in the functional group abundances. If the model is not good at all at predicting next year observations, your elasticities might be a bit theoretical.
2) on lines 61-62, you wrote that the model allocates equal prior probability to bottom-up versus top-down control. You should explain what you mean here (that is what are the probabilities involved - is it the case because you have two groups of predators and two groups of producers, with identical priors for the different alphas defining the interactions). How would you do it with for example 3 groups of producers and 2 groups of predators?
3) l. 80 "makes including all known sources of uncertainty essential". Do you really think you have achieved this? It is not just a matter of including a process in name (eg measurement error).
4) Table 1: you define the lognormal in terms of centre and scale but usually it is defined in terms of the mean and variance of log(Y). Please explain.
5) l. 398 ff: could you give the time needed to run the MCMC chains? Useful information for others.

Reviewer 1 ·

Basic reporting

Meets all standards

Experimental design

Meets all standards

Validity of the findings

Meets all standards

Additional comments

In this manuscript , the authors apply state-of-the-art analytical methods to address an extremely important issue in fisheries ecology – structural stability of functional groups in response to exploitation in a fishery ecosystem. Elasticity analyses are used very effectively throughout the paper to address this key issue and to convey main results. I strongly support publication of this paper following some revision It is a nice complement to a series of important papers on the Bay of Biscay ecosystem published over the last decade or more. Some general points are provided below.
1) The concept of elasticity is very clearly defined in the paper. As noted above, I think it is very valuable. My sense though is that some readers may not be familiar with its use and its relationship to the perhaps more familiar concept of sensitivity analysis. There may be some merit in stating that it is a non-dimensional sensitivity analysis using proportional changes to permit direct comparison of the relative effect of changes in different parameters measured in different units (or something to that effect).
2) It may be worth mentioning that the general lessons from this analysis are (perhaps not surprisingly) consistent with that of Bell et al. (2014) in their analysis of the Celtic Sea using the same species groups. So although the geographic area is not precisely the same, a different time period (1987-2004) was covered, and somewhat different model structure (with explicit incorporation of fishing pressure), treatment of fleets, and analytical method (non-Bayesian) was used, many of the same messages emerge where direct comparison is possible.
3) Figure 5 does not seem to be described in the text. I think this is a pivotal figure and it deserves careful evaluation. In examining this figure, I was left wondering about how best to deal with the issue of the elasticity of the density-independent parameter, particularly in relation to fishing pressure. If f > r the population rate of change will of course be negative. It wasn’t entirely clear to me if the points made on negative values of r (lines 245-255) capture the issue mentioned above. The treatment in the lines cited above seems to deal just with ri. But I’m wondering about ri-fi. since resilience to exploitation will depend on the relative magnitudes of ri and fi.. Put another way, doesn’t interpretation of Figure 5 for the different functional groups involve conditioning on ri to understand the fishing elasticities shown in this figure? In any event, a section needs to be added to the paper to deal with this figure if it is to remain.
4) The paragraph in lines 534-543 needs work. The second sentence seems to be missing something. The following two sentences seem to be a non–sequitur. The first of these two lines points to uncertainty in detecting a bottom up effect. In the following sentence in which we expect an explanation, an explanation for the difficulty in detecting top-down effects is given.

Reviewer 2 ·

Basic reporting

Clear and unambiguous, professional English used throughout.
> The English level is undoubtedly professional.

The article must be written in English and must use clear, unambiguous, technically correct text. The article must conform to professional standards of courtesy and expression.
> Same comment as above.

Literature references, sufficient field background/context provided.
> The provided litterature is suffisient.

The article should include sufficient introduction and background to demonstrate how the work fits into the broader field of knowledge. Relevant prior literature should be appropriately referenced.
> The added value of that study is not clear enougth and must be shown in a more explicit way given the litterature.

Professional article structure, figures, tables. Raw data shared.
> Undoubtedly professional.

The structure of the article should conform to an acceptable format of ‘standard sections’ (see our Instructions for Authors for our suggested format). Significant departures in structure should be made only if they significantly improve clarity or conform to a discipline-specific custom.
> The structure seems in accordance with the suggested format of PeerJ.

Figures should be relevant to the content of the article, of sufficient resolution, and appropriately described and labeled.
> Figures/tables are nice.

All appropriate raw data have been made available in accordance with our Data Sharing policy.
> Raw data are provided by the authors.

Experimental design

Original primary research within Aims and Scope of the journal.
> The content of that manuscript is in line with those of PeerJ.

Research question well defined, relevant & meaningful. It is stated how research fills an identified knowledge gap. The submission should clearly define the research question, which must be relevant and meaningful. The knowledge gap being investigated should be identified, and statements should be made as to how the study contributes to filling that gap.
> Research questions are not well defined. Same comment for the knowledge gap.

Rigorous investigation performed to a high technical & ethical standard. The investigation must have been conducted rigorously and to a high technical standard. The research must have been conducted in conformity with the prevailing ethical standards in the field.
> Numerical analyses are particularly developed and sophisticated.

Methods described with sufficient detail & information to replicate. Methods should be described with sufficient information to be reproducible by another investigator.
> Methods are not enough explicit and thus not replicable, which my main concern for that manuscript.

Validity of the findings

Decisions are not made based on any subjective determination of impact, degree of advance, novelty, being of interest to only a niche audience, etc. Replication experiments are encouraged (provided the rationale for the replication, and how it adds value to the literature, is clearly described); however, we do not allow the ‘pointless’ repetition of well known, widely accepted results.
> The results obtained are not ‘pointless’ repetition of well known results but suffer from an explicit description an explanation.

Data is robust, statistically sound, & controlled. The data on which the conclusions are based must be provided or made available in an acceptable discipline-specific repository. The data should be robust, statistically sound, and controlled.
> The data used seem robust.

Conclusions are well stated, linked to original research question & limited to supporting results. The conclusions should be appropriately stated, should be connected to the original question investigated, and should be limited to those supported by the results.
> Conclusions are particularly hard to read and understand. However, I am convinced that manuscript could benefit from a simplification of the writing.

Speculation is welcome, but should be identified as such.
> To my point of view, any speculation was made here.

Additional comments

The manuscript submitted by Hosack and Trenkel is impressive by his huge amount of data mining but is extremely hard to read and understand on a biological point of view. To convince a broader panel of scientists, I highly suggest to render the manuscript more simple, especially concerning statistical investigations. The manuscript could benefit from a more simple description of numerical methods. Moreover, the link between numerical investigations and ecological considerations could be improved: the reader often have to switch from statistical to ecological paragraph, which is pretty disruptive and then render the overall story very hard to understand. In all cases I’m convinced the paper could be useful but clarification is needed, like for example enumerating clear questions at the end of the introduction part. The hierarchy between objectives in terms of priority must be clearly stated. As an example, in the abstract, several objectives are mentioned : ”The potential impacts of historical fisheries management on functional groups are estimated”, “The importance of density dependence, such as top-down versus bottom--up ecosystem regulation, is also investigated”, “The long term response of the ecosystem to a range of management scenarios is also investigated”. We also can read “We then propose to evaluate ecosystem response to perturbations by investigating elasticities » at the end of the introduction, which creates some confusion on the real aim of that study. My main suggestion is to provide a manuscript written in a way that other scientists could reproduce same investigations on 'their' own datasets.

---

## Round 0.2 · accepted · Accept

Thanks for making a thorough revision of the paper and providing detailed and constructive answers to the comments made on the first version of the paper. I will definitely use this paper for my own work.